# LoveDA: A Remote Sensing Land-Cover Dataset for Domain Adaptive Semantic Segmentation

**Junjue Wang,**[*] **Zhuo Zheng,**[*] **Ailong Ma, Xiaoyan Lu, Yanfei Zhong**[†]
State Key Laboratory of Information Engineering in Surveying, Mapping, and Remote Sensing
Wuhan University, Wuhan 430074, China
`{kingdrone,zhengzhuo,maailong007,luxiaoyan,zhongyanfei}@whu.edu.cn`

## Abstract

Deep learning approaches have shown promising results in remote sensing high spatial resolution (HSR) land-cover mapping. However, urban and rural scenes can show completely different geographical landscapes, and the inadequate generalizability of these algorithms hinders city-level or national-level mapping. Most of the existing HSR land-cover datasets mainly promote the research of learning semantic representation, thereby ignoring the model transferability. In this paper, we introduce the Land-cOVEr Domain Adaptive semantic segmentation (LoveDA) dataset to advance semantic and transferable learning. The LoveDA dataset contains 5987 HSR images with 166768 annotated objects from three different cities. Compared to the existing datasets, the LoveDA dataset encompasses two domains (urban and rural), which brings considerable challenges due to the: 1) multi-scale objects; 2) complex background samples; and 3) inconsistent class distributions. The LoveDA dataset is suitable for both land-cover semantic segmentation and unsupervised domain adaptation (UDA) tasks. Accordingly, we benchmarked the LoveDA dataset on eleven semantic segmentation methods and eight UDA methods. Some exploratory studies including multi-scale architectures and strategies, additional background supervision, and pseudo-label analysis were also carried out to address these challenges. The code and data are available at https://github.com/Junjue-Wang/LoveDA.

## 1 Introduction

With the continuous development of society and economy, the human living environment is gradually being differentiated, and can be divided into urban and rural zones [8]. High spatial resolution (HSR) remote sensing technology can help us to better understand the geographical and ecological environment. Specifically, land-cover semantic segmentation in remote sensing is aimed at determining the land-cover type at every image pixel. The existing HSR land-cover datasets such as the Gaofen Image Dataset (GID) [38], DeepGlobe [9], Zeebruges [24], and Zurich Summer [43] contain large-scale images with pixel-wise annotations, thus promoting the development of fully convolutional networks (FCNs) in the field of remote sensing [46, 49]. However, these datasets are designed only for semantic segmentation, and they ignore the diverse styles among geographic areas. For urban and rural areas, in particular, the manifestation of the land cover is completely different, in the class distributions, object scales, and pixel spectra. In order to improve the model generalizability for large-scale land-cover mapping, appropriate datasets are required.

In this paper, we introduce an HSR dataset for Land-cOVEr Domain Adaptive semantic segmentation (LoveDA) for use in two challenging tasks: semantic segmentation and UDA. Compared with the

---

[*]Equal contribution.
[†]Corresponding author.

35th Conference on Neural Information Processing Systems (NeurIPS 2021) Track on Datasets and Benchmarks.

UDA datasets [22, 39] that use simulated images, the LoveDA dataset contains real urban and rural remote sensing images. Exploring the use of deep transfer learning methods on this dataset will be a meaningful way to promote large-scale land-cover mapping. The major characteristics of this dataset are summarized as follows: **1) Multi-scale objects.** The HSR images were collected from 18 complex urban and rural scenes, covering three different cities in China. The objects in the same category are in completely different geographical landscapes in the different scenes, which increases the scale variation. **2) Complex background samples.** The remote sensing semantic segmentation task is always faced with the complex background samples (i.e., land-cover objects that are not of interest) [29, 54], which is particularly the case in the LoveDA dataset. The high-resolution and different complex scenes bring more rich details as well as larger intra-class variance for the background samples. **3) Inconsistent class distributions.** The urban and rural scenes have different class distributions. The urban scenes with high population densities contain lots of artificial objects such as buildings and roads. In contrast, the rural scenes include more natural elements, such as forest and water. Compared with UDA datasets [30, 42] in general computer vision, the LoveDA dataset focuses on the style differences of the geographical environments. The inconsistent class distributions pose a special challenge for the UDA task.

As the LoveDA dataset was built with two tasks in mind, both advanced semantic segmentation and UDA methods were evaluated. Several exploratory experiments were also conducted to solve the particular challenges inherent in this dataset, and to inspire further research. A stronger representational architecture and UDA method are needed to jointly promote large-scale land cover mapping.

## 2   Related Work

Table 1: Comparison between LoveDA and the main land-cover semantic segmentation datasets.

| Image level | Resolution (m) | Dataset | Year | Sensor | Area (km$^2$) | Classes | Image width | Images | Task SS | Task UDA |
|---|---|---|---|---|---|---|---|---|---|---|
| Meter level | 10 | LandCoverNet [1] | 2020 | Sentinel-2 | 30000 | 7 | 256 | 1980 | ✓ | |
| | 4 | GID [38] | 2020 | GF-2 | 75900 | 5 | 4800~6300 | 150 | ✓ | |
| Sub-meter level | 0.25~0.5 | LandCover.ai [2] | 2020 | Airborne | 216.27 | 3 | 4200~9500 | 41 | ✓ | |
| | 0.6 | Zurich Summer [43] | 2015 | QuickBird | 9.37 | 8 | 622~1830 | 20 | ✓ | |
| | 0.5 | DeepGlobe [9] | 2018 | WorldView-2 | 1716.9 | 7 | 2448 | 1146 | ✓ | |
| | 0.05 | Zeebruges [24] | 2018 | Airborne | 1.75 | 8 | 10000 | 7 | ✓ | |
| | 0.05 | ISPRS Potsdam 3 | 2013 | Airborne | 3.42 | 6 | 6000 | 38 | ✓ | |
| | 0.09 | ISPRS Vaihingen 4 | 2013 | Airborne | 1.38 | 6 | 1887~3816 | 33 | ✓ | |
| | 0.07 | AIRS [6] | 2019 | Airborne | 475 | 2 | 10000 | 1047 | ✓ | |
| | 0.5 | SpaceNet [41] | 2017 | WorldView-2 | 2544 | 2 | 406~439 | 6000 | ✓ | |
| | 0.3 | LoveDA (Ours) | 2021 | Spaceborne | 536.15 | 7 | 1024 | 5987 | ✓ | ✓ |

The abbreviations are: SS – semantic segmentation, UDA – unsupervised domain adaptation.

### 2.1   Land-cover semantic segmentation datasets

Land-cover semantic segmentation, as a long-standing research topic, has been widely explored over the past decades. The early research relied on low- and medium-resolution datasets, such as MCD12Q1 [33], the National Land Cover Database (NLCD) [14], GlobeLand30 [15], LandCoverNet [1], etc. However, these studies all focused on large-scale mapping and analysis from a macro-level. With the advancement of remote sensing technology, massive HSR images are now being obtained on a daily basis from both spaceborne and airborne platforms. Due to the advantages of the clear geometrical structure and fine texture, HSR land-cover datasets are tailored for specific scenes at a micro-level. As is shown in Table 1, datasets such as ISPRS Potsdam [3], ISPRS Vaihingen [4], Zurich Summer [43], and Zeebruges [24] are designed for urban parsing. These datasets only contain a small number of annotated images and cover limited areas. In contrast, DeepGlobe [9] and LandCover.ai [2] focus on rural areas with a larger scale, in which the homogeneous areas contain few man-made structures. The GID dataset[38] was collected with Gaofen-2 satellite from different cities in China. Although LandCoverNet and GID datasets contain both urban and rural areas, the geo-locations of these released images are private. Therefore, the urban and rural areas are not able to be divided. In addition, the identifications of cities in released GID images have been already removed so it is

---

[3]http://www2.isprs.org/commissions/comm3/wg4/2d-sem-label-potsdam.html
[4]http://www2.isprs.org/commissions/comm3/wg4/2d-sem-label-vaihingen.html

hard to perform UDA tasks. Considering limited coverage and annotation cost, the existing HSR datasets mainly promote the research of improving land-cover segmentation accuracy, ignoring its transferability. Compared with land-cover datasets, the iSAID dataset[48] focuses on key objects semantic segmentation. The different study objects bring different challenges for different remote sensing tasks.

These HSR land-cover datasets have all promoted the development of semantic segmentation, and many variants of FCNs [19] have been evaluated [7, 10, 11, 46]. Recently, some UDA methods have been developed from the combination of two public datasets [50]. However, directly utilizing combined datasets may result in two problems: 1) Insufficient common categories. Different datasets are designed for different purposes, and the insufficient common categories limit further exploration. 2) Inconsistent annotation granularity. The different spatial resolutions and labeling styles lead to different annotation granularities, which can result in unreliable conclusions. Compared with existing datasets, LoveDA dataset encompasses two domains (urban and rural), representing a novel UDA task for land-cover mapping.

### 2.2 Unsupervised domain adaptation

For natural images, UDA is aimed at transferring a model trained on the source domain to the target domain. Some conventional image classification studies [20, 34, 40] have directly minimized the discrepancy of the feature distributions to extract domain-invariant features. The recent works have mainly proceeded in two directions, i.e., adversarial training and self-training.

**Adversarial training**. In adversarial training, the architecture includes a feature extractor and a discriminator. The extractor aims to learn domain-invariant features, while the discriminator attempts to distinguish these features. For semantic segmentation, Tsai et al. [39] considered the semantic outputs containing spatial similarities between the different domains, and adapted the structured output space for segmentation (AdaptSeg) with adversarial learning. Luo et al. [22] introduced a category-level adversarial network (CLAN) to align each class with an adaptive adversarial loss. Differing from the binary discriminators, Wang et al. [44] proposed a fine-grained adversarial learning framework for domain adaptive semantic segmentation (FADA), aligning the class-level features. From the aspect of structure, the transferable normalization (TransNorm) method [47] was proposed to enhance the transferability of the FCN-based feature extractors. All these advanced adversarial learning methods were implemented on the LoveDA dataset for evaluation.

**Self-training**. Self-training involves alternately generating pseudo-labels on the target data and fine-tuning the model. Recently, the self-training UDA methods have focused on improving the quality of the pseudo-labels [51, 57]. Lian et al. [18] designed the self-motivated pyramid curriculum (PyCDA) to observe the target properties, and fused multi-scale features. Zou et al. [56] proposed a class-balanced self-training (CBST) strategy to sample pseudo-labels, thus avoiding the dominance of the large classes. Mei et al. [25] used an instance adaptive self-training (IAST) selector for sample balance. In addition to testing these self-training methods on the LoveDA dataset, we also performed the pseudo-label analysis for the CBST.

**UDA in the remote sensing community**. The early UDA methods focused on scene classification tasks [21, 28]. Recently, adversarial training [13, 35] and self-training [38] have been studied for UDA land-cover semantic segmentation. These methods follow the general UDA approach in the computer vision field, with some improvements. However, with only the public datasets, the advancement of the UDA algorithms has been limited by the insufficient shared categories and the inconsistent annotation granularity. To this end, the LoveDA dataset is proposed for a more challenging benchmark, promoting future research of remote sensing UDA algorithms and applications.

## 3 Dataset Description

### 3.1 Image Distribution and Division

The LoveDA dataset was constructed using 0.3 m images obtained from Nanjing, Changzhou and Wuhan in July 2016, totally covering $536.15\mathrm{km}^2$ (Figure 1). The historical images were obtained from the Google Earth platform. As each research area has its own planning strategy, the urban-rural ratio is inconsistent [52].

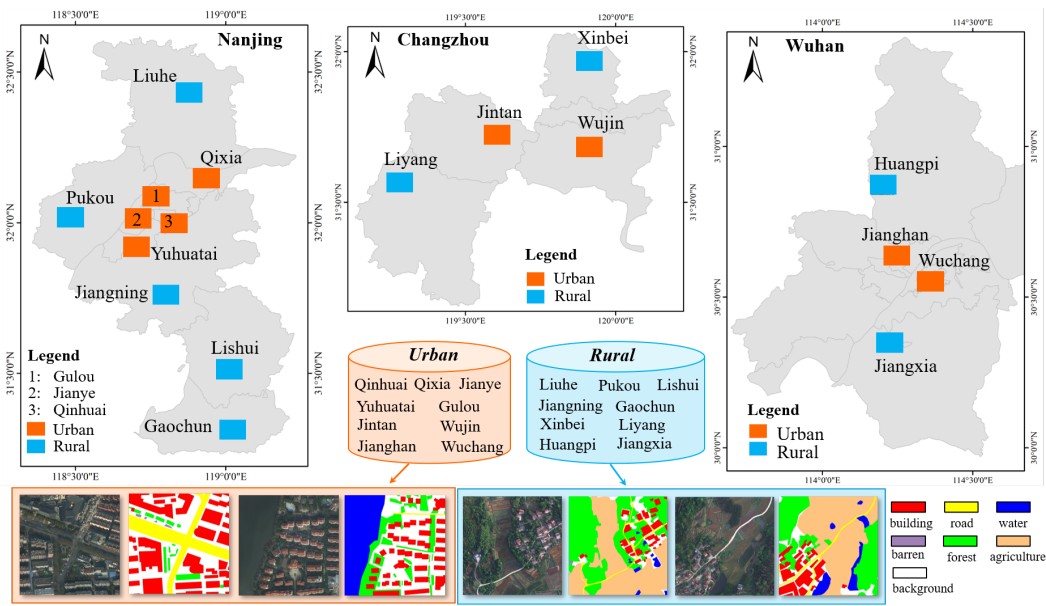

Figure 1: Overview of the dataset distribution. The images were collected from Nanjing, Changzhou, and Wuhan cities, covering 18 different administrative districts.

Data from the rural and urban areas were collected referring to the "Urban and Rural Division Code" issued by the National Bureau of Statistics. There are nine urban areas selected from different economically developed districts, which are all densely populated ($> 1000 \; \mathrm{people/km^2}$) [52]. The other nine rural areas were selected from undeveloped districts. The spatial resolution is 0.3 m, with red, green, and blue bands. After geometric registration and pre-processing, each area is covered by $1024 \times 1024$ images, without overlap. Considering Tobler's First Law, i.e., everything is related to everything else, but near things are more related than distant things [37], the training, validation, and test sets were split so that they were spatially independent (Figure 1), thus enhancing the difference between the split sets. There are two tasks that can be evaluated on the LoveDA dataset: **1) Semantic segmentation**. There are eight areas for training, and the others are for validation and testing. The training, validation, and test sets cover both urban and rural areas.**2) Unsupervised domain adaptation**. The UDA process considers two cross-domain adaptation sub-tasks: *a) Urban → Rural.* The images from the Qinhuai, Qixia, Jianghan, and Gulou areas are included in the source training set. The images from Liuhe and Huangpi are included in the validation set. The Jiangning, Xinbei, and Liyang images included in the test set. The *Oracle* setting is designed to test the upper limit of accuracy in a single domain [31]. Hence, the training images were collected from the Pukou, Lishui, Gaochun, and Jiangxia areas. *b) Rural → Urban.* The images from the Pukou, Lishui, Gaochun, and Jiangxia areas are included in the source training set. The images from Yuhuatai and Jintan are used for the validation set. The Jiangye, Wuchang, and Wujin images are used for the test set. In the *Oracle* setting, the training images cover the Qinhuai, Qixia, Jianghan, and Gulou areas.

With the division of these images, a comprehensive annotation pipeline was adopted, including professional annotators and strict inspection procedures [48]. Further details of the data division and annotation can be found in §A.1.

## 3.2 Statistics for LoveDA

Some statistics of the LoveDA dataset are analyzed in this section. With the collection of public HSR land-cover datasets, the number of labeled objects and pixels has been counted. As is shown in the Figure 2(a), our proposed LoveDA dataset contains the largest number of labeled pixels as well as land-cover objects, which shows the advantage in data diversity. There are a lot of buildings because urban scenes have large populations (Figure 2(b)). As is shown in Figure 2(c), the background class contains the most pixels with complex samples [29, 54]. The **complex background samples** have larger intra-class variance in the complex scenes and cause serious false alarms.

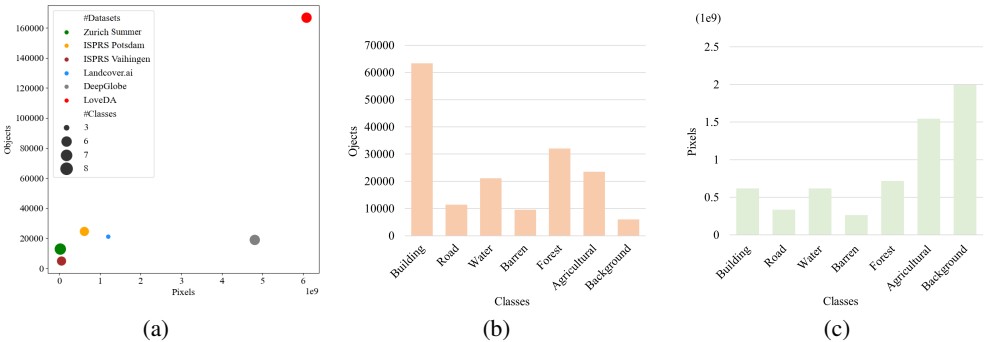

(a)             (b)             (c)

Figure 2: Statistics for the pixels and objects in LoveDA dataset. (a) Number of objects vs. number of pixels. The radius of the circles represents the number of classes. (b) Histogram of the number of objects for each class. (c) Histogram of the number of pixels for each class.

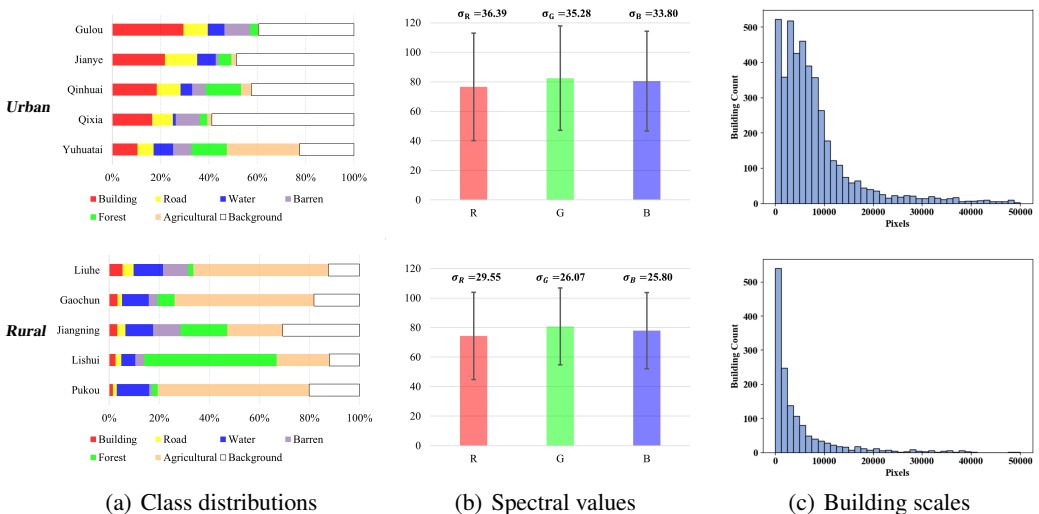

(a) Class distributions      (b) Spectral values      (c) Building scales

Figure 3: Statistics for the urban and rural scenes in Nanjing City. (a) Class distribution. (b) Spectral statistics. The mean and standard deviation ($\sigma$) for 5 urban and 5 rural areas are reported. (c) Distribution of the building sizes. The Jianye (urban) and Lishui (rural) scenes are reported.

### 3.3 Differences Between Urban and Rural Scenes

During the process of urbanization, cities differentiate into rural and urban forms. In this section, we list the main differences, which reveal the meaning and challenges of the remote sensing UDA task. For the Nanjing City, the main differences come from the shape, layout, scale, spectra, and class distribution. As is shown in Figure 1, the buildings in the urban area are neatly arranged, with various shapes, while the buildings in the rural area are disordered, with simpler shapes. The roads are wide in the urban scenes. In contrast, the roads are narrow in the rural scenes. Water is often presented in the form of large-scale rivers or lakes in the urban scenes, while small-scale ponds and ditches are common in the rural scenes. The agricultural is found in the gaps between the buildings in the urban scenes, but occurs in a large-scale and continuously distributed form in the rural scenes.

For the class distribution, spectra, and scale, the related statistics are reported in Figure 3. The urban areas always contain more man-made objects such as buildings and roads due to their high population density (Figure 3(a)). In contrast, the rural areas have more agricultural land. The **inconsistent class distributions** between the urban and rural scenes increases the difficulty of model generalization. For the spectral statistics, the mean values are similar (Figure 3(b)). Because of the large-scale homogeneous geographical areas, such as agriculture and water, the rural images have lower standard

deviations. As is shown in Figure 3(c), most of the buildings have relatively small scales in the rural areas, representing the "long tail" phenomenon. However, the buildings in the urban scenes have a larger size variance. Scale differences also exist in the other categories, as shown in Figure 1. The **multi-scale objects** require the models to have multi-scale capture capabilities. When faced with large-scale land cover mapping tasks, the differences between urban and rural scenes bring new challenges to the model transferability.

## 4 Experiments

### 4.1 Semantic Segmentation

For the semantic segmentation task, the general architectures as well as their variants, and particularly those most often used in remote sensing, were tested on the LoveDA dataset. Specifically, the selected networks were: UNet[32], UNet++[55], LinkNet[3], DeepLabV3+[5], PSPNet[53], FCN8S[19], PAN[17], Semantic-FPN[16], HRNet[45], FarSeg[54], and FactSeg[23]. Following the common practice[19, 45], we use the intersection over union (IoU) to report the semantic segmentation accuracy. With respect to the IoU for each class, the mIoU represents the mean of the IoUs over all the categories. The inference speed is reported with a single $512 \times 512$ input (repeated $500$ times), using frames per second (FPS).

Table 2: Semantic segmentation results obtained on the `Test` set of LoveDA.

| Method | Backbone | IoU per category (%) | | | | | | | mIoU (%) | Speed (FPS) |
|---|---|---|---|---|---|---|---|---|---|---|
| | | Background | Building | Road | Water | Barren | Forest | Agriculture | | |
| FCN8S [19] | VGG16 | 42.60 | 49.51 | 48.05 | 73.09 | 11.84 | 43.49 | 58.30 | 46.69 | 86.02 |
| DeepLabV3+ [5] | ResNet50 | 42.97 | 50.88 | 52.02 | 74.36 | 10.40 | 44.21 | 58.53 | 47.62 | 75.33 |
| PAN [17] | ResNet50 | 43.04 | 51.34 | 50.93 | 74.77 | 10.03 | 42.19 | 57.65 | 47.13 | 61.09 |
| UNet [32] | ResNet50 | 43.06 | 52.74 | 52.78 | 73.08 | 10.33 | 43.05 | 59.87 | 47.84 | 71.35 |
| UNet++ [55] | ResNet50 | 42.85 | 52.58 | 52.82 | 74.51 | 11.42 | 44.42 | 58.80 | 48.20 | 27.22 |
| Semantic-FPN [16] | ResNet50 | 42.93 | 51.53 | 53.43 | 74.67 | 11.21 | 44.62 | 58.68 | 48.15 | 73.98 |
| PSPNet [53] | ResNet50 | 44.40 | 52.13 | 53.52 | 76.50 | 9.73 | 44.07 | 57.85 | 48.31 | 74.81 |
| LinkNet [3] | ResNet50 | 43.61 | 52.07 | 52.53 | 76.85 | 12.16 | 45.05 | 57.25 | 48.50 | 67.01 |
| FarSeg [54] | ResNet50 | 43.09 | 51.48 | 53.85 | 76.61 | 9.78 | 43.33 | 58.90 | 48.15 | 66.99 |
| FactSeg [23] | ResNet50 | 42.60 | 53.63 | 52.79 | **76.94** | **16.20** | 42.92 | 57.50 | 48.94 | 65.58 |
| HRNet [45] | W32 | **44.61** | **55.34** | **57.42** | 73.96 | 11.07 | **45.25** | **60.88** | **49.79** | 16.74 |

Table 3: Multi-Scale augmentation during Training and Testing (MSTrTe).

| Method | mIoU(%) | | |
|---|---|---|---|
| | Baseline | +MSTr | +MSTrTe |
| DeepLabV3+ | 47.62 | 49.97 | 51.18 |
| UNet | 48.00 | 50.21 | 51.13 |
| SFPN | 48.15 | 50.80 | 51.82 |
| HRNet | 49.79 | 51.51 | 52.14 |

**Implementation details**. The data splits followed the Table 8 in §A.1. During the training, we used the Stochastic Gradient Descent (SGD) optimizer with a momentum of $0.9$ and a weight decay of $10^{-4}$. The learning rate was initially set to $0.01$, and a 'poly' schedule with power $0.9$ was applied. The number of training iterations was set to $15k$ with a batch size of 16. For the data augmentation, $512 \times 512$ patches were randomly cropped from the raw images, with random mirroring and rotation. The backbones used in all the networks were pre-trained on ImageNet.

**Multi-scale architectures and strategies**. As ground objects show considerable scale variance, especially in complex scenes (§3.3), we have analyzed the multi-scale architectures and strategies. There are three noticeable observations from Table 2: 1) UNet++ outperforms UNet due to its nested cross-scale connections between different scales. 2) Among the different fusion strategies, UNet++, Semantic-FPN, LinkNet and HRNet outperform DeepLabV3+. This demonstrates that the cross-layer fusion works better than the in-module fusion. 3) HRNet outperforms the other methods, due to its sophisticated architecture, where the features are repeatedly exchanged across different scales. As is shown in Table 3, multi-scale augmentation (with $scale = \{0.5, 0.75, 1.0, 1.25, 1.5, 1.75\}$) was

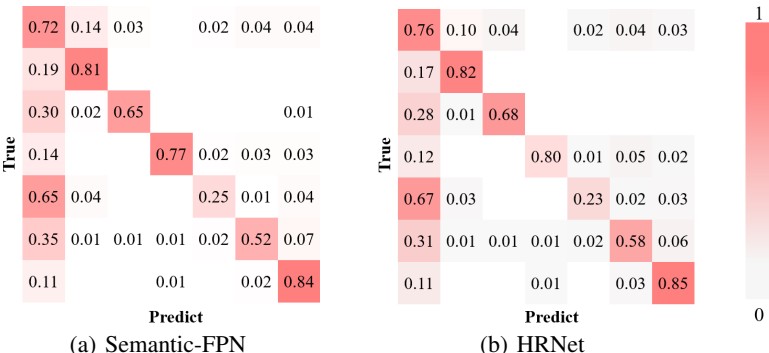

(a) Semantic-FPN  (b) HRNet

Figure 4: Representative confusion matrices for the semantic segmentation experiments.

conducted during the training (MSTr), significantly improving the performance of different methods. In the implementation, the multi-scale inference adopts multi-scale inputs and ensembles the rescaled multiple outputs using a simple mean function. With further use in the testing process, all methods were further improved. As for multi-scale fusion, hierarchical multi-scale architecture search [46] may also become an effective solution.

**Additional background supervision**. The complex background samples cause serious false alarms in HRS imagery semantic segmentation [12, 54]. As is shown in Figure 4, the confusion matrices show that lots of objects were misclassified into background, which is consistent with our analysis in §3.2. Based on Semantic-FPN, we designed the additional background supervision to address this problem. Dice loss [26] and binary cross-entropy loss were utilized with the corresponding modulation factors. We calculated the total loss as: $L_{total} = L_{ce} + \alpha L_{bce} + \beta L_{dice}$, where $L_{ce}$ denotes the original cross-entropy loss. Table 4 and Table 5 additionally report the precision (P), recall (R) and F1-score (F1) of the background class with varying modulation factors. Besides, the standard deviations are reported after 3 runs. Table 4 shows that the addition of binary cross-entropy loss improves the background accuracy and the overall performance. The combination of $L_{dice}$ and $L_{bce}$ performs well because they optimize the background class from different directions. In the future, the spatial attention mechanism [27] may improve the background with adaptive weights.

Table 4: Varied $\alpha$ for $L_{bce}$

| $\alpha$ | Background | | | mIoU (%) |
| --- | --- | --- | --- | --- |
| | P (%) | R (%) | F1(%) | |
| 0 | 55.46 | 61.01 | 59.86 | $48.15 \pm 0.17$ |
| 0.2 | 57.70 | 63.36 | 60.39 | $48.50 \pm 0.13$ |
| 0.5 | 56.92 | 65.86 | 61.06 | $\mathbf{48.85} \pm 0.15$ |
| 0.7 | 57.73 | 64.62 | 61.98 | $\mathbf{48.74} \pm 0.19$ |
| 0.9 | 57.30 | 64.05 | 60.48 | $48.26 \pm 0.14$ |
| 1.0 | 58.43 | 62.64 | 60.46 | $48.14 \pm 0.18$ |

Table 5: Varied $\beta$ for $L_{dice}$ (w. optimal $\alpha$)

| $\beta$ | $\alpha$ | Background | | | mIoU (%) |
| --- | --- | --- | --- | --- | --- |
| | | P (%) | R (%) | F1(%) | |
| 0.2 | 0.5 | 56.68 | 64.82 | 60.47 | $48.97 \pm 0.16$ |
| 0.5 | 0.5 | 56.88 | 65.16 | 60.96 | $49.23 \pm 0.09$ |
| 0.7 | 0.5 | 57.13 | 65.31 | 60.93 | $49.68 \pm 0.14$ |
| 0.2 | 0.7 | 56.91 | 66.03 | 61.13 | $49.69 \pm 0.17$ |
| 0.5 | 0.7 | 57.14 | 66.21 | 61.34 | $\mathbf{50.08} \pm 0.15$ |
| 0.7 | 0.7 | 56.68 | 65.52 | 60.78 | $49.48 \pm 0.13$ |

**Visualization**. Some representative results are shown in Figure 5. With the shallow backbone (VGG16), FCN8S can hardly recognize the road due to its lack of feature extraction capability. The other methods which utilize deep layers can produce better results. Because of the disorderly arrangement and varied scales, the edges of the buildings are hard to extract accurately. Some small-scale objects such as buildings and scattered trees are easy to miss. In contrast, water class achieves higher accuracies for all methods. This because water have strong spectral homogeneity and low intra-class variance [38]. The forest is easy to misclassify into agriculture because these classes have similar spectra. Because of the high-resolution retention and multi-scale fusion, HRNet produces the best visualization result, especially in the details.

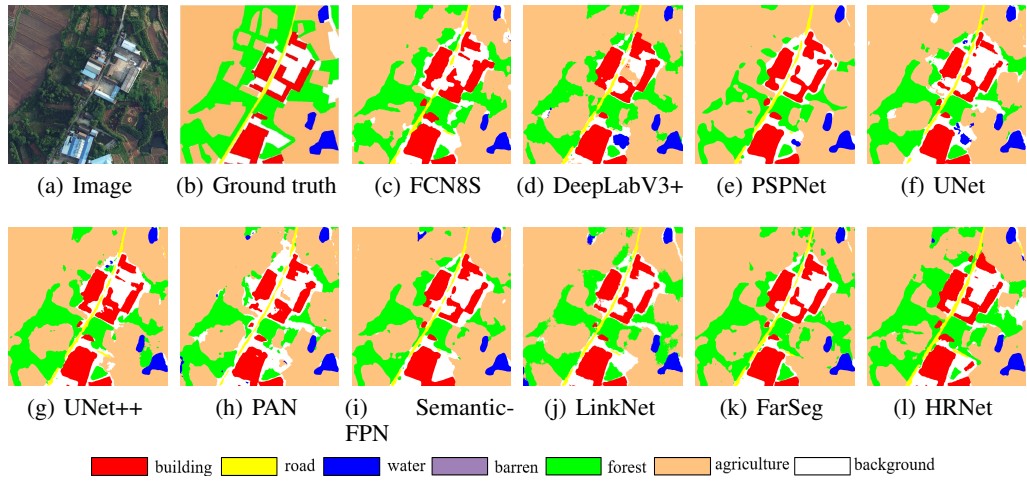

| | | | | | |
|---|---|---|---|---|---|
| (a) Image | (b) Ground truth | (c) FCN8S | (d) DeepLabV3+ | (e) PSPNet | (f) UNet |
| (g) UNet++ | (h) PAN | (i) Semantic-FPN | (j) LinkNet | (k) FarSeg | (l) HRNet |

■ building  ■ road  ■ water  ■ barren  ■ forest  ■ agriculture  □ background

Figure 5: Semantic segmentation results on images from the LoveDA `Test` set in the Liuhe (**Rural**) area. Some small-scale objects such as buildings and scattered trees are hard to recognize. The forest and agricultural classes are easy to misclassify due to their similar spectra.

## 4.2 Unsupervised Domain Adaptation

The advanced UDA methods were evaluated on the LoveDA dataset. In addition to the original metric-based approach of MCD [40], two mainstream UDA approaches were tested, i.e., adversarial training (AdaptSeg [39], CLAN [22], TransNorm [47], FADA [44]) and self-training (CBST [56], PyCDA [18], IAST [25]).

Table 6: Unsupervised domain adaptation results obtained on the `Test` set of the LoveDA dataset.

| Domain | Method | Type | IoU (%) | | | | | | | mIoU(%) |
|---|---|---|---|---|---|---|---|---|---|---|
| | | | Background | Building | Road | Water | Barren | Forest | Agriculture | |
| | Oracle | - | 48.18 | 52.14 | 56.81 | 85.72 | 12.34 | 36.70 | 35.66 | 46.79 |
| **Rural** ↓ Urban | Source only | - | 43.30 | 25.63 | 12.70 | 76.22 | 12.52 | 23.34 | 25.14 | 31.27 |
| | MCD [40] | - | 43.60 | 15.37 | 11.98 | 79.07 | 14.13 | 33.08 | 23.47 | 31.53 |
| | AdaptSeg [39] | AT | 42.35 | 23.73 | 15.61 | 81.95 | 13.62 | 28.70 | 22.05 | 32.68 |
| | FADA [44] | AT | 43.89 | 12.62 | 12.76 | 80.37 | 12.70 | 32.76 | 24.79 | 31.41 |
| | CLAN [22] | AT | 43.41 | 25.42 | 13.75 | 79.25 | 13.71 | 30.44 | 25.80 | 33.11 |
| | TransNorm [47] | AT | 38.37 | 5.04 | 3.75 | 80.83 | 14.19 | 33.99 | 17.91 | 27.73 |
| | PyCDA [18] | ST | 38.04 | 35.86 | **45.51** | 74.87 | 7.71 | **40.39** | 11.39 | 36.25 |
| | CBST [56] | ST | 48.37 | **46.10** | 35.79 | 80.05 | 19.18 | 29.69 | 30.05 | **41.32** |
| | IAST [25] | ST | **48.57** | 31.51 | 28.73 | **86.01** | **20.29** | 31.77 | **36.50** | 40.48 |
| | Oracle | - | 37.18 | 52.74 | 43.74 | 65.89 | 11.47 | 45.78 | 62.91 | 45.67 |
| **Urban** ↓ Rural | Source only | - | 24.16 | 37.02 | **32.56** | 49.42 | 14.00 | 29.34 | 35.65 | 31.74 |
| | MCD [40] | - | 25.61 | 44.27 | 31.28 | 44.78 | 13.74 | 33.83 | 25.98 | 31.36 |
| | AdaptSeg [39] | AT | 26.89 | 40.53 | 30.65 | 50.09 | 16.97 | 32.51 | 28.25 | 32.27 |
| | FADA [44] | AT | 24.39 | 32.97 | 25.61 | 47.59 | 15.34 | 34.35 | 20.29 | 28.65 |
| | CLAN [22] | AT | 22.93 | 44.78 | 25.99 | 46.81 | 10.54 | 37.21 | 24.45 | 30.39 |
| | TransNorm [47] | AT | 19.39 | 36.30 | 22.04 | 36.68 | 14.00 | **40.62** | 3.30 | 24.62 |
| | PyCDA [18] | ST | 12.36 | 38.11 | 20.45 | 57.16 | **18.32** | 36.71 | 41.90 | 32.14 |
| | CBST [56] | ST | 25.06 | 44.02 | 23.79 | 50.48 | 8.33 | 39.16 | 49.65 | 34.36 |
| | IAST [25] | ST | **29.97** | **49.48** | 28.29 | **64.49** | 2.13 | 33.36 | **61.37** | **38.44** |

The abbreviations are: AT – adversarial training methods. ST – self-training methods.

**Implementation details.** All the UDA methods adopted the same feature extractor and discriminator, following the common practice [22, 39, 44]. Specifically, DeepLabV2 [4] with ResNet50 was utilized as the extractor, and the discriminator was constructed by fully convolutional layers [39]. For the adversarial training (AT), the classification and discriminator learning rates were set to $5 \times 10^{-3}$ and $10^{-4}$, respectively. The Adam optimizer was used for the discriminator with the momentum of $0.9$ and $0.99$. The number of training iterations was set to $10k$, with a batch size of 16. The eight source images and eight target images were alternatively input. The other settings are the same in the

semantic segmentation. and the learning schedule is the same as in semantic segmentation settings. For the self-training (ST), the classification learning rate was set to $10^{-2}$. Full implementation details are provided in the §A.4.

**Benchmark results.** As is shown in Table 6, the *Oracle setting* obtains the best overall performances. However, DeepLabV2 has lost its effectiveness due to the domain divergence, referring to the result of *Source only* setting. In the **Rural** → Urban experiments, the accuracies of artificial classes (building and road) drop more than natural classes (forest and agricultural). Because of the inconsistent class distribution, the **Urban** → Rural experiments show the opposite results. The transfer learning methods relatively improve the model transferability. Noticeably, TransNorm obtains the lowest mIoUs. This is because the source and target images were obtained by the same sensor, and their spectral statistics are similar (Figure 3(2)). These rural and urban domains require similar normalization weights, so that the adaptive normalization can lead to optimization conflicts (more analysis are provided in §A.6). The ST methods achieve better performances because they address the class imbalance problem with pseudo-label generation.

**Inconsistent class distributions.** It is noticeable to find that the ST methods surpass AT methods in cross-domain adaptation experiments. We conclude that the main reason for this is the extremely inconsistent class distribution (Figure 3(a)). The rural scenes only contain a few artificial samples and large-scale natural objects. In contrast, the urban scenes have a mixture of buildings and roads with few natural objects. The AT methods cannot address this difficulty, so that they report lower accuracies. However, differing from the AT methods, the ST methods generate pseudo-labels on the target images. With the addition of diverse target samples, the class distribution divergence is eliminated during the training. Overall, the ST methods show more potential in the UDA land-cover semantic segmentation task. In **Urban** → Rural experiments, all UDA methods show negative transfer effects for the road class. Hence, more tailored UDA methods are worth exploring faced with these special challenges.

**Visualization.** The qualitative results for the **Rural** → Urban experiments are shown in Figure 6. The *Oracle* result successfully recognizes the buildings and roads, and is the closest to the ground truth. According to the Table 2, it can be further improved by using a more robust backbone. The ST methods (j)–(l) produce better results than AT methods (f)–(i), but there is still much room for improvement. The large-scale mapping visualizations are provided in §A.7.

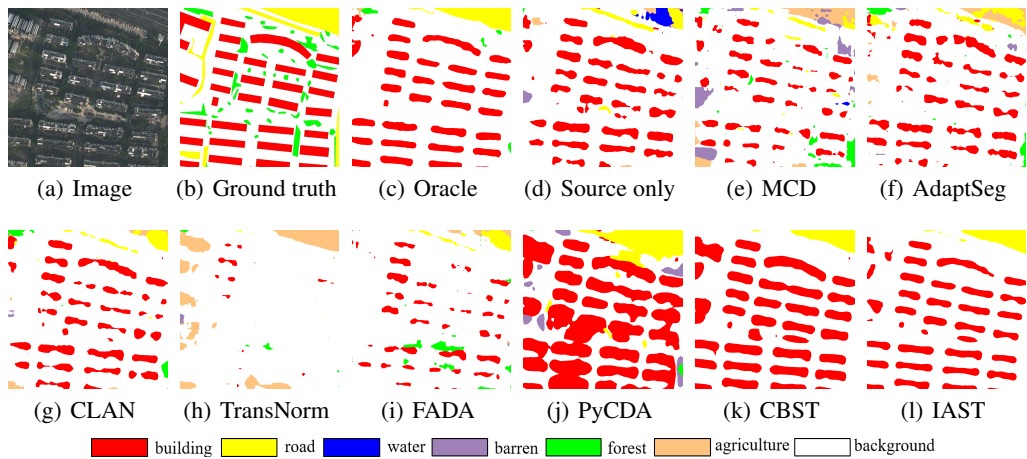

Figure 6: Visual results for the **Rural** → Urban experiments. (f)–(i) and (j)–(l) were obtained from the AT and ST methods, respectively. The ST methods produce better results than the AT methods.

**Pseudo-label analysis for CBST.** As pseudo samples are important for addressing inconsistent class distribution problem, we varied the target class proportion in CBST, which is a hyper-parameter controlling the number of pseudo samples. The mean F1-score (mF1) and mIoU are reported in Table 7. Without pseudo-label learning ($t = 0$), the model degenerated into *Source only* setting and achieved low accuracy. The optimal range of $t$ is relatively large ($0.05 \leq t \leq 0.5$), which proves that it is not sensitive to the remote sensing UDA task.

Table 7: Varied $p$ for target class proportion (**Rural** $\rightarrow$ Urban)

| $t$ | 0. | 0.01 | 0.05 | 0.1 | 0.5 | 0.7 | 0.9 | 1.0 |
|---|---|---|---|---|---|---|---|---|
| mF1(%) | 46.81 | 45.24 | 48.50 | 50.93 | **56.30** | 51.23 | 51.03 | 49.43 |
| mIoU(%) | 32.94 | 32.18 | 34.46 | 36.84 | **41.32** | 37.12 | 37.02 | 35.47 |

## 5 Conclusion

The differences between urban and rural scenes limit the generalization of deep learning approaches in land-cover mapping. In order to address this problem, we built an HSR dataset for Land-cOVEr Domain Adaptive semantic segmentation (LoveDA). The LoveDA dataset reflects three challenges in large-scale remote sensing mapping, including multi-scale objects, complex background samples, and inconsistent class distributions. The state-of-the-art methods were evaluated on the LoveDA dataset, revealing the challenges of LoveDA. In addition, multi-scale architectures and strategies, additional background supervision and pseudo-label analysis were conducted to find alternative ways to address these challenges.

## 6 Broader Impact

This work offers a free and open dataset with the purpose of advancing land-cover semantic segmentation in the area of remote sensing. We also provide two benchmarked tasks with three considerable challenges. This will allow other researchers to easily build on this work and create new and enhanced capabilities. The authors do not foresee any negative societal impacts of this work. A potential positive societal impact may arise from the development of generalizable models that can produce large-scale high-spatial-resolution land-cover mapping accurately. This could help to reduce the manpower and material resource consumption of surveying and mapping.

## 7 Acknowledgments

This work was supported by National Key Research and Development Program of China under Grant No. 2017YFB0504202, National Natural Science Foundation of China under Grant Nos. 41771385, 41801267, and the China Postdoctoral Science Foundation under Grant 2017M622522. This work was supported by the Nanjing Bureau of Surveying and Mapping.

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
