# OpenReview forum: "LoveDA: A Remote Sensing Land-Cover Dataset for Domain Adaptive Semantic Segmentation"
_NeurIPS.cc/2021/Track/Datasets_and_Benchmarks/Round2 — NeurIPS 2021 Datasets and Benchmarks Track (Round 2)_

### Official Review · Reviewer_RMNt · 2021-09-21
**The description of the data is unclear to me.**

**Rating:** 6
**Confidence:** 4
**Correctness:** The data source is unclear.
**Clarity:** Yes, but some descriptions are unclea…

**Strengths:**

- a large number of remote sensing images with high resolution are provided.
- the challenge of the dataset is described for deep learning methods.
- the conducted experiments are sufficient, and the analysis of multi-scale architectures and strategies is very nice.

**Weaknesses:**

- the authors described that their data are collected by airborne sensors, but it is still not detailed enough to me.
- they only described the annotation process in the datasheet, no collection process is described.
- the authors claimed that they added two cities (Changzhou, Wuhan) to increase the data diversity, but they did not explain the difference among the three cities.
- I am also concerned about the privacy and security of cities. To my knowledge, the permission of the Chinese government is necessary even without geo-locations, if the authors want to use or open the high-resolution images. The authors claimed that they obtained the permission of Nanjing Bureau of Surveying and Mapping, how about Changzhou and Wuhan?
- It lacks a document about the dataset. The mask images look completely black, how to use them? And many HSR images have large black regions, they are padded for uniform sizes? So the noise is involved, the authors should describe the impact.
- In the 27 line of Introduction, the description " these datasets are designed for single-domain" may be incorrect. [8] and [33] included diverse areas, and [33] also studied transfer learning on high-resolution remote sensing images.
- the motivation of transfer learning between rural and urban is unclear to me. The rural and urban scenes are different, but is it necessary to study transfer learning between these two scenes in practice? or is this task only designed for improving the transfer learning methods?

**Additional Feedback:**

- In 1(b) of checklist, "more effective methods need to be designed" is not the limitation of the introduced dataset.
- I think that the privacy and security problems should be described in 1(c) of checklist.
- more analysis of experiments results can be added to make contributions to the deep learning community.

**Documentation:**

The description of data collection and the instruction of the dataset are lacking.

**Ethics:**

I am concerned about the privacy and security of Changzhou and Wuhan.

**Relation To Prior Work:**

The author should clarify: the prior works are all single-domain? and no domain adaptation is studied?

**Summary And Contributions:**

This paper introduced a large-scale dataset with annotated objects from three different cities. The statistics of the dataset are described and the challenges are also explained, solid experiments of the existing methods are conducted on multiple tasks. However, some descriptions of the data is unclear to me, please see the weakness for details. I would like to change my rating according to the results of the discussion.

Their contributions include:
- 5972 aerial images with 166768 annotated objects.
- solid experiments of the existing methods on semantic segmentation and domain adaptation.

---

> ### Author Response · Authors · 2021-09-25
> **Response to Reviewer RMNt (Part3)**
>
> **Q5: "In the 27 line of Introduction, the description " these datasets are designed for single-domain" may be incorrect. [8] and [33] included diverse areas, and [33] also studied transfer learning on high-resolution remote sensing images."**
>
> Thanks for your careful suggestion. We have clarified this description (P.1) as follows:
> *"However, these datasets mainly promote the semantic representation of deep learning models, ignoring the transferable challenges caused by the differences between rural and urban scenes."*
>
> **Q6: "the motivation of transfer learning between rural and urban is unclear to me. The rural and urban scenes are different, but is it necessary to study transfer learning between these two scenes in practice? or is this task only designed for improving the transfer learning methods?"**
>
> Thanks for your careful suggestion. Due to the high cost of HSR remote sensing image annotations, our research found that many local surveying and mapping departments only have small historical labels, covering small regions. In the practical mapping application, requirements for the model generalization are naturally raised. Especially the differences between rural and urban scenes limit the generalization. To this end, we have proposed the LoveDA dataset, promoting large-scale mapping for surveying and mapping. Besides, the LoveDA dataset provides a solid benchmark for the research of UDA methods.
>
> The LoveDA is not only designed for transfer learning methods but also semantic segmentation methods. These two tasks have been individually benchmarked with ten semantic segmentation and eight domain adaptation models.
>
> **Q7: "In 1(b) of checklist, "more effective methods need to be designed" is not the limitation of the introduced dataset.
> I think that the privacy and security problems should be described in 1(c) of checklist.
> more analysis of experiments results can be added to make contributions to the deep learning community."**
>
> Thanks for your constructive comment. In the checklist, we have removed the limitation “more effective methods need to be designed” and the data source is also described in the 1(c) of the checklist as follows:
>
> *“The images in Nanjing City were shared under permission of Nanjing Bureau of Surveying and Mapping. The extended images covering Changzhou and Wuhan City were obtained from Google Earth Engine (GEE) and these are open-source images. All the geographical coordinates were further removed through a decryption processing.”*
>
> Besides, we will continue to explore new methods on this dataset to make contributions to the deep learning community.
>
> We hope our response can address your concerns. We kindly ask you to reconsider your final score. Please let us know if you have
> any further questions.

---

> ### Author Response · Authors · 2021-09-25
> **Response to Reviewer RMNt (Part2)**
>
> **Q3: "I am also concerned about the privacy and security of cities. To my knowledge, the permission of the Chinese government is necessary even without geo-locations, if the authors want to use or open the high-resolution images. The authors claimed that they obtained the permission of Nanjing Bureau of Surveying and Mapping, how about Changzhou and Wuhan?"**
>
> Sorry for this confusion.
> The extended images covering Changzhou and Wuhan City were obtained from Google Earth Engine (GEE) and these are open-source spaceborne images. We have clarified the data source in the revised version as
> follows:
>
> *“The LoveDA dataset was constructed using 0.3 m images obtained from Nanjing, Changzhou, and Wuhan in July 2016, covering 536.15 $km^2$ (Figure 1). The images in Nanjing were captured with a Leica DMC airborne digital camera, and the spaceborne images in Changzhou and Wuhan were collected from the Google Earth Engine (GEE).”*
>
> **Q4: "It lacks a document about the dataset. The mask images look completely black, how to use them? And many HSR images have large black regions, they are padded for uniform sizes? So, the noise is involved, the authors should describe the impact."**
>
> Thanks for your careful suggestion. The values of mask images refer to the land-cover types: 0 - ignore, 1 - background, 2 - building, 3 - road, 4 - water, 5 - barren, 6 - forest, 7 - agriculture. These values are low so the masks look dark. The dataset API has been provided in our code to help construct the pipeline. Besides, a rendering API will be added to our code for visualization. The black regions in HSR images are caused by the geometric correction. We have set the corresponding mask pixel as ignore values referred to Pascal VOC, ISPRS Potsdam semantic segmentation datasets. During the training, the model optimization can ignore these ignore pixels by setting the ignore index = 0 in the loss function. This was implemented in our provided PyTorch code.

---

> ### Author Response · Authors · 2021-09-25
> **Response to Reviewer RMNt (Part1)**
>
> Thanks for your thoughtful review. We hope our response can address your concerns.
>
> **Q1:"the authors described that their data are collected by airborne sensors, but it is still not detailed enough to me. they only described the annotation process in the datasheet, no collection process is described."**
>
> Thanks for your careful suggestion. In Round1, the images in Nanjing City were obtained through project cooperation with the Nanjing Bureau of Surveying Mapping. Under their permission, these images were shared through decryption processing. To increase the data diversity in Round2, Changzhou and Wuhan cities were added and the historical images were downloaded from the Google Earth, which is open-source. The collection process was added in the revised paper (P.4) as follows:
>
> *"The LoveDA dataset was constructed using 0.3 m images obtained from Nanjing, Changzhou and Wuhan in July 2016, covering 536.15 $\rm{km^2}$ (Figure 1).The images in Nanjing were captured with a Leica DMC airborne digital camera, and the images in Changzhou and Wuhan were collected from the Google Earth."*
>
> **Q2: "the authors claimed that they added two cities (Changzhou, Wuhan) to increase the data diversity, but they did not explain the difference among the three cities."**
>
> Thanks for your constructive comment. Based on the comments in Round1, we expanded the study area and data volume, so we expanded two new cities, each with its own characteristics [56]. As each research area has its own planning strategy, the urban-rural ratio is inconsistent. For example, Nanjing and Changzhou both belong to Jiangsu Province but their developments are different. The urban-rural ratios of Nanjing and Changzhou are 1.78 and 0.37 [49]. Wuhan and Nanjing are both provincial capital cities, but due to different development plans and geographical environments, the ratio of Wuhan is only 0.20 [49].
> Therefore, the three research areas can better reflect the generalization of the UDA methods in different domains. At the same time, in different study areas, the proportions of land cover between classes are different.
>
> In summary, the urban-rural ratios and the proportions of land cover types vary in different research areas, which provides rich diversity and heterogeneity for the dataset, and brings new challenges to the transfer learning algorithm.
> We also added the description in the revised paper (P.4) as follows:
>
> “As each research area has its own planning strategy, the urban-rural ratio is inconsistent. Referred to [49], the urban-rural ratios of Nanjing, Changzhou, and Wuhan are 1.78, 0.37, and 0.20 respectively.”
>
> **Reference**
> - [49] H. Zhao. National urban population and construction land in 2016 (by cities). China Statistics Press, 2016.
> - [56] Zhong Y, Su Y, Wu S, et al. Open-source data-driven urban land-use mapping integrating point-line-polygon semantic objects: A case study of Chinese cities[J]. Remote Sensing of Environment, 2020, 247: 111838.

---

> > ### Comment · Reviewer_RMNt · 2021-09-26
> > **What is the resolution of Changzhou and Wuhan from GEE?**
> >
> > Thank you for the explanations.
> >
> > What is the resolution of Changzhou and Wuhan's images from GEE? Is it the same as Nanjing 0.3m?

---

> > > ### Author Response · Authors · 2021-09-26
> > > **Thank you for your prompt reply**
> > >
> > > Thank you for your prompt reply.
> > >
> > > Yes, in order to keep the same resolution as Nanjing, we downloaded the historical images at 0.3m.
> > >
> > > Please let us know if you have any further questions.

---

> > > > ### Comment · Reviewer_RMNt · 2021-09-27
> > > > **I change my rating from 5 to 6**
> > > >
> > > > Thank you for the reply.
> > > >
> > > > I change my rating from 5 to 6.

---

### Official Review · Reviewer_HU8B · 2021-09-21
**A new remote sensing dataset that contains semantic segmentation and domain adaptation tasks**

**Rating:** 7
**Confidence:** 3
**Correctness:** The technical aspects of the work see…
**Clarity:** The paper is structurally well-writte…

**Strengths:**

1. The introduced dataset focuses on the domain differences between urban and rural scenes and collected images from representative regions. This is the first multi-domain dataset for remote sensing images and shall promote the related research areas. And I agree their response, that a combined dataset is not a good choice.
2. The analysis of the dataset statistics is very detailed. Through the analysis, the differences between the two domains are very obvious, thus convinces the high demand of increasing the model generability.
3. The authors conduct a comprehensive study of baseline models on both two tasks and list both the qualitative and quantitative results.

**Weaknesses:**

1. The fonts of the figures are still quite small.
2. Even though the dataset has been extended with images from more cities, the coverage area is still limited. These cities all lie in the middle and lower reaches of the Yangtze River. A better choice is covering more regions, each region sample fewer images.

**Additional Feedback:**

None

**Documentation:**

No URLs are linked to the paper, thus currently the documentation is not provided.

**Ethics:**

The authors claim to follow the "Urban and Rural Division Code" issued by the National Bureau of Statistics.
It is released under the permission of the Nanjing Bureau of Surveying and Mapping.

**Relation To Prior Work:**

The authors demonstrate a good comparison to other similar dataset, highlighting their strengths against others.

**Summary And Contributions:**

This paper introduces a large-scale dataset LoveDA for semantic segmentation and domain adaptation with HSR images. Unlike existing datasets, the proposed dataset focuses on improving model transferability between urban and rural scenes. The dataset shows obvious style differences and class inconsistencies between the two kinds of scenes. The HSR images are collected from 18 complex urban and rural scenes in three different cities in China. The final dataset contains 5,927 images with a resolution of 0.3 m. Several semantic segmentation and domain adaptation models are tested on the proposed dataset.

---

> ### Author Response · Authors · 2021-09-26
> **Response to Reviewer HU8B**
>
> We would first like to thank the reviewer for his/her positive support and for the time reviewing our work. Below you will find our responses to your comments.
>
> **Q1: "The fonts of the figures are still quite small."**
>
> Thanks for your careful suggestion. We have revised the fonts in the figures carefully, which have been enlarged by four levels.
> The revised paper has been successfully updated.
>
> **Q2: "Even though the dataset has been extended with images from more cities, the coverage area is still limited. These cities all lie in the middle and lower reaches of the Yangtze River. A better choice is covering more regions, each region sample fewer images."**
>
> Thanks for your constructive suggestion. Through the Round2 of dataset expansion, it verified that our data collection and annotation pipelines are effective. Refer to your suggestion, we will expand it to more cities in the future.
>
>
> **Q3:"No URLs are linked to the paper, thus currently the documentation is not provided."**
>
> Thanks for your careful suggestion. We have already provided the documents including the datasheet, data division in the supplementary material at: https://openreview.net/attachment?id=bLBIbVaGDu&name=supplementary_material. The link of data and code were also shared at: https://openreview.net/forum?id=bLBIbVaGDu&noteId=2WpGlesA_cr
>
> We hope our response can address your concerns, and please let us know if you have any further questions.

---

### Official Review · Reviewer_8Fht · 2021-09-23
**Valuable HSR land-cover mapping dataset**

**Rating:** 6
**Confidence:** 3

**Strengths:**

- Even though the dataset is not the best in all the different characteristics (km2 area covered, classes, image width, ...) it achieves a good compromise overall.
- Providing domain labels is also a very strong point to be able to test methods and strategies that are to be deployed later in real-world scenarios.
- Useful statistics about the data are also provided.
- Extensive benchmarking by using state-of-the-art models.


**Weaknesses:**

1) If any, I'd ask the authors if there are no other datasets to compare to. After a quick look I've found some that could be related, and potentially included in Table 1.
- Airbus Ship
- SpaceNet
- iSAID
- AIRS https://www.airs-dataset.com/
- ...
iSAID (2019) being particularly interesting having 15 categories, 655,451 instances, 2,806 images, and image widths 800~13,000.

If not relevant, it'd be great that the authors explicitly state why they are not. Doesn't it fit the semantic segmentation task?

2) Not illustrating top performances (ideally using the same metric as in this paper, i.e. mIOU) on other datasets. This would support the "challeangability" of the proposed dataset compared to other existing ones

**Additional Feedback:**

I would gladly have decided "7: Good paper, accept" instead of "6: Marginally above acceptance threshold" if Table 1 have provided a stronger overview and comparison w.r.t. to other existing datasets. I'd also like to know why other datasets (especially, iSAID) are not considered.

**Clarity:**

- Table 1 would be much easier to interpret if the years in which the differents datasets were released were included as an additional column.
- Although it is included in the Data sheet, the paper lacks some information regarding the data: (a) is the number of images captured in each of the 18 areas fairly balanced - yes, I know it form the Data sheet -, (b) is the number of images and/or instances balanced across domains?, etc. Please, revise the data sheet to see if any relevant information was not included in the paper.
- I'd have expected at least sentence explaining the multi-scale inference in MSTrTe.

# Minors

- Line 34: "using" -> "use"
- Line 142: "Figure 2(a), Our" -> "Figure 2(a), our"
- Line 153: "simple" -> "simpler"
- Figure 2b and 2c: keep the position of the "Background" bar the same for the two subfigures.
- Figure 3's caption: "for 10 areas are reported" is a bit confusing although, perhaps "for 5 urban and 5 rural areas are reported" would be clearer.
- Line 201: "Semnatic-FPN" -> "Semantic-FPN"
- Figure 5's caption: "Visual results" -> "Semantic segmentation results" and "regconize" -> "recognize".
- Line 261: actually, mP and mR are NOT in Table 7.
- Table 7: "porprotion" -> "proportion"

**Correctness:**

Yes. Both of these are true:
- The dataset is constructed in a sound way.
- Benchmarking of the dataset using state-of-the-art methods is properly designed and performed.

**Documentation:**

Data sheet and data division documents where provided. The information in the Data sheet addresses all those questions successfully.

**Ethics:**

No concerns, given the images cover wide areas and the images are NOT provided along with coordinates.

**Relation To Prior Work:**

I understand the value of the proposed dataset is its size but, even more important, the inclusion of domain labels, which makes the problem much more realistic and enables the proper evaluation of future aerial image semantic segmentation approaches. Those methods could fall short by not even thinking about the huge variability across countries, cities, urban/rural, etc. This kind of dataset did not exist, thus making it quite valuable.

**Summary And Contributions:**

The authors propose a new dataset for high spatial resolution land-cover mapping, a problem that is often posed as a semantic segmentation task. The dataset is fairly large compared to other datasets reported in the paper, multiple classes of objects are annotated through the images with polygonal masks, and an additional label tells whether the images belong to the urban or rural domains. The latter characteristic makes the dataset distinct compared to others, which are often collected in only one or few locations and/or multiple locations but without this information. The paper provides also an extensive benchmarking for semantic segmentation and domain adaptation tasks using state-of-the-art methods demonstrating the challenges of the dataset.

---

> ### Author Response · Authors · 2021-09-25
> **Response to Reviewer 8Fht (Part3)**
>
> **Q4: ''Although it is included in the Data sheet, the paper lacks some information regarding the data: (a) is the number of images captured in each of the 18 areas fairly balanced - yes, I know it form the Data sheet -, (b) is the number of images and/or instances balanced across domains?, etc. Please, revise the data sheet to see if any relevant information was not included in the paper. ''**
>
> Thanks for your careful suggestion. As for (a), due to space limitations, we have clarified a number of images captured in each of the 18 areas in Section A.2 (P.16) in the Appendix. As for (b), the domain differences have been analyzed in Section 3.3, clarifying that the pixel-level class distributions are imbalanced across domains. Besides, the instance-level differences were analyzed in Section A.3 (P.16) and summarized as follows:
>
> |  Type            |     Background    |     Building    |     Road    |     Water    |     Barren    |     Forest    |     Agricultural    |
> |--------------|-------------------|-----------------|-------------|--------------|---------------|---------------|---------------------|
> |     Urban    |     3010          |     44056       |     5068    |     5132     |     3428      |     18665     |     4864            |
> |     Rural    |     3334          |     21136       |     4259    |     13698    |     3538      |     11233     |     16043           |
>
> The corresponding analysis have been added as follows (P.16) *'''The differences between urban and rural areas at instance-level are shown below (also see histograms in Figure 7). Similar with the pixel analysis in Section 3.3, the instances across domains are imbalanced. Specifically, the urban areas have more buildings and fewer instances of agricultural land. The rural areas have more instances of agricultural land. This also highlights the inconsistent class distribution problem between different domains.''*
>
> **Q5: ''I'd have expected at least sentence explaining the multi-scale inference in MSTrTe.''**
>
> Thanks for your careful suggestion. The multi-scale inference was explained in the revised paper (P.7) as follows:
> *“In the implementation, the multi-scale inference adopts multi-scale inputs and ensembles the rescaled multiple outputs using a simple mean function.”*
>
> **Q6: ''I would gladly have decided "7: Good paper, accept" instead of "6: Marginally above acceptance threshold" if Table 1 have provided a stronger overview and comparison w.r.t. to other existing datasets. I'd also like to know why other datasets (especially, iSAID) are not considered.''**
>
> Thank you so much for considering improving your score. To provide a stronger overview, we have added SpaceNet and AIRS datasets comparison and extend a new column for released years in Table I (P.2). The similarities and differences compared with iSAID are clarified as follows. The iSAID and our proposed LoveDA are both for semantic segmentation but for **different remote sensing scientific problems.** **1) Different study objects:** The iSAID dataset is designed for key objects such as ships, cars, planes that are movable. These objects can not reflect the characteristics of the geographical environment. However, the LoveDA studies the fixed land-cover objects that reflect the biophysical material over the surface of the earth. **2) Different challenges:** The iSAID dataset promotes the multi-scale problem due to their object-scale variation [46]. In contrast, because the land-cover objects are influenced by the geographical environment and different scenes manifest different land-cover distributions. The LoveDA dataset not only promotes the multi-scale segmentation problem but also promotes UDA task between different scenes in remote sensing.
>
> Referred to your good suggestion, we also consider the iSAID dataset and revised our paper (P.3):
>
> *''Compared with land-cover datasets, the iSAID dataset [46] focuses on key objects semantic segmentation. The different study objects bring different challenges for different remote sensing tasks.''*
>
> **Reference**
> - [46] Waqas Zamir S, Arora A, Gupta A, et al. isaid: A large-scale dataset for instance segmentation in aerial images[C]//Proceedings of the IEEE/CVF Conference on Computer Vision and Pattern Recognition Workshops. 2019: 28-37.
>
> Besides, the minor description errors have been carefully revised. The modifications have been highlighted with red color text in the uploaded version. We hope our response can address your concerns, and please let us know if you have any further questions.

---

> ### Author Response · Authors · 2021-09-25
> **Response to Reviewer 8Fht (Part2)**
>
> **Q2: ''Not illustrating top performances (ideally using the same metric as in this paper, i.e. mIOU) on other datasets. This would support the "challeangability" of the proposed dataset compared to other existing ones.''**
>
> Thanks for your careful suggestion. Considering the different classification systems, divisions and experimental settings, the compared top performances seem not convincing to some extent. But we also believe that this could support the "challeangability" of the proposed dataset. To this end, we have added this compared table in Appendix A.8 (P.20) as follows:
>
> *''By investigating the current researches, the top performances on different datasets have been reported in Table 9. The advanced method (HRNet) only achieved the lowest performance on the LoveDA dataset, showing the difficulty of this dataset''*
>
> |     Dataset             |     Top mIoU (%)    |
> |-------------------------|---------------------|
> |     GID[44]              |     93.54           |
> |     DeepGlobe[35]        |     52.24           |
> |     ISPRS Potsdam[26]    |     82.38           |
> |     ISPRS Vaihingen[26]        |     79.76           |
> |     SpaceNet[9]         |     50.75           |
> |     LoveDA                |     49.79           |
>
> **Reference**
> - [44] Wang J, Zhong Y, Zheng Z, et al. RSNet: The search for remote sensing deep neural networks in recognition tasks[J]. IEEE Transactions on Geoscience and Remote Sensing, 2020, 59(3): 2520-2534.
> - [35] Tian C, Li C, Shi J. Dense fusion classmate network for land cover classification[C]//Proceedings of the IEEE Conference on Computer Vision and Pattern Recognition Workshops. 2018: 192-196.
> - [26] Mou L, Hua Y, Zhu X X. A relation-augmented fully convolutional network for semantic segmentation in aerial scenes[C]//Proceedings of the IEEE/CVF Conference on Computer Vision and Pattern Recognition. 2019: 12416-12425.
> - [9] Coulter L, Hall T, Guzman L, et al. Satellite Image Building Detection using U-Net Convolutional Neural Network[J].
>
> **Q3: ''Table 1 would be much easier to interpret if the years in which the different datasets were released were included as an additional column.''**
>
> Thanks for your careful suggestion. The released years have been added as an additional column in Table I (P.2).

---

> ### Author Response · Authors · 2021-09-25
> **Response to Reviewer 8Fht (Part1)**
>
> We would first like to thank the reviewer for his/her positive support and for the time reviewing our work. Below you will find our responses to your comments.
>
> **Q1: ''If any, I'd ask the authors if there are no other datasets to compare to. After a quick look I've found some that could be related, and potentially included in Table 1. Airbus Ship, SpaceNet, iSAID, AIRS https://www.airs-dataset.com/, iSAID (2019). If not relevant, it'd be great that the authors explicitly state why they are not. Doesn't it fit the semantic segmentation task?''**
>
> Thanks for your careful suggestion. These datasets and our proposed LoveDA are both for semantic segmentation but for different remote sensing scientific problems. **1) Different study objects:** The iSAID and Airbus Ship datasets are designed for key objects such as ships, cars, planes that are movable. These objects can not reflect the characteristics of the geographical environment. However, the LoveDA studies the fixed land-cover objects that reflect the biophysical material over the surface of the earth. The SpaceNet and AIRS datasets only focus the building extraction but the LoveDA dataset focuses on the general land-cover types including buildings, water, forest, etc. **2) Different challenges:** The Airbus Ship, SpaceNet, iSAID, and AIRS datasets promote the multi-scale problem due to their object-scale variation [46]. In contrast, because the land-cover objects are influenced by the geographical environment and different scenes manifest different land-cover distributions. The LoveDA dataset not only promotes the multi-scale segmentation problem but also promotes UDA task between different scenes in remote sensing.
>
> Thank you for pointing out this important concern. The comparative details with SpaceNet and AIRS datasets have been added in Table I (P.2) as follows:
>
> |     Dataset     | Year |      Sensor        | Resolution (m) | Area (km2) | Classes |
> |:---------------:|:----:|:--------------------:|:--------------:|:----------:|:-------:|
> |   LandCoverNet  | 2020 |      Sentinel-2      |       10       |    30000   |    7    |
> |       GID       | 2020 |         GF-2         |        4       |    75900   |    5    |
> |   LandCover.ai  | 2020 |       Airborne       |    0.25~0.5    |   216.27   |    3    |
> |  Zurich Summer  | 2015 |       QuickBird      |       0.6      |    9.37    |    8    |
> |    DeepGlobe    | 2018 |      WorldView-2     |       0.5      |   1716.9   |    7    |
> |    Zeebruges    | 2018 |       Airborne       |      0.05      |    1.75    |    8    |
> |  ISPRS Potsdam  | 2013 |       Airborne       |      0.05      |    3.42    |    6    |
> | ISPRS Vaihingen | 2013 |       Airborne       |      0.09      |    1.38    |    6    |
> |       AIRS      | 2019 |       Airborne       |      0.07      |     475    |    2    |
> |     SpaceNet    | 2017 |      WorldView-2     |       0.5      |    2544    |    2    |
> |   LoveDA(Ours)  | 2021 | Airborne, Spaceborne |       0.3      |   536.15   |    7    |
>
> **Reference**
> - [46] Waqas Zamir S, Arora A, Gupta A, et al. isaid: A large-scale dataset for instance segmentation in aerial images[C]//Proceedings of the IEEE/CVF Conference on Computer Vision and Pattern Recognition Workshops. 2019: 28-37.

---

### Author Response · Authors · 2021-09-29
**Updates in the manuscript**

We thank all the reviewers for their helpful comments and are pleased that our work has been well-received overall.

According to every comment from all the reviewers, we have carefully revised our paper and update the 10-page-version paper.
The major modifications are listed as follows:
- Table 1 has been extended to provide a stronger overview and comparison to other existing datasets. (8Fht)
- Table 9 has been added to report top performances on different datasets. (8Fht)
- Figure 7 has been added to show the differences across domains at the instance level. (8Fht)
- The comparison with iSAID has been clarified in P.3. (8Fht)
- The fonts in the figures have been enlarged by four levels. (HU8B)
- The collection and security of images have been clarified in P.4 and P.13. (RMNt)

Hope our responses and the updated version can dismiss your concerns, and please let us know if you have any further questions.

---

### Decision · Program_Chairs · 2021-10-11

**Decision:**

Accept

**Comment:**

All reviewers agree to accept the paper. After reading the paper, AC does not see flaws to contradict the reviewers' opinions.